# ReAlnet: Achieving More Human Brain-Like Vision via Human Neural Representational Alignment

**Zitong Lu**[*]**& Julie D. Golomb**
Department of Psychology
The Ohio State University
Columbus, OH 43210, USA
`{lu.2637, golomb.9}@osu.edu`

**Yile Wang**
Department of Neuroscience
The University of Texas at Dallas
Richardson, TX 75080, USA
`yile.wang@utdallas.edu`

## Abstract

Despite the remarkable strides made in artificial intelligence, current object recognition models still lag behind in emulating the mechanism of visual information processing in human brains. Recent studies have highlighted the potential of using neural data to mimic brain processing; however, these often reply on invasive neural recordings from non-human subjects, leaving a critical gap in our understanding of human visual perception and the development of more human brain-like vision models. Addressing this gap, we present, for the first time, 'Re(presentational)Al(ignment)net', a vision model aligned with human brain activity based on non-invasive EEG recordings, demonstrating a significantly higher similarity to human brain representations. Our innovative image-to-brain multi-layer encoding alignment framework not only optimizes multiple layers of the model, marking a substantial leap in neural alignment, but also enables the model to efficiently learn and mimic human brain's visual representational patterns across object categories and different neural data modalities. Furthermore, we discover that alignment with human brain representations improves the model's adversarial robustness. Our findings suggest that ReAlnet sets a new precedent in the field, bridging the gap between artificial and human vision, and paving the way for more brain-like artificial intelligence systems.

## 1 Introduction

While current vision models in artificial intelligence (AI) are advanced, they still fall short of capturing the full complexity and adaptability inherent in the human brain's information processing. Deep convolutional neural networks (DCNNs) have reached a performance level in object recognition that rivals human capabilities Lecun et al. (2015), and many studies have identified similarities in the hierarchical structure between DCNNs and the ventral visual stream Cichy et al. (2016); Güçlü & van Gerven (2015); Kietzmann et al. (2019); Lu & Golomb (2023a); Yamins et al. (2014). However, the alignment between DCNNs and human neural representations remains deeply inadequate, whether compared with human electroencephalography (EEG) or functional magnetic resonance imaging (fMRI) data. Enhancing the resemblance between visual models and the human brain has become a critical concern for both computer scientists and neuroscientists. From a computer vision perspective, brain-inspired models often exhibit higher robustness and generalization, crucial for realizing true brain-like intelligence; meanwhile, from a cognitive neuroscience perspective, models that more closely mirror brain representations can significantly aid in our exploration of the brain's visual processing mechanisms.

Given these challenges and limitations, the pivotal question arises is **how we can leverage our understanding of the human brain to enhance current AI vision models.** Conventional approaches have limitations in emulating the complexity of the human brain's visual information processing, even with increased model depth and layers Rajalingham et al. (2018). This limitation has prompted

---
[*]Corresponding author

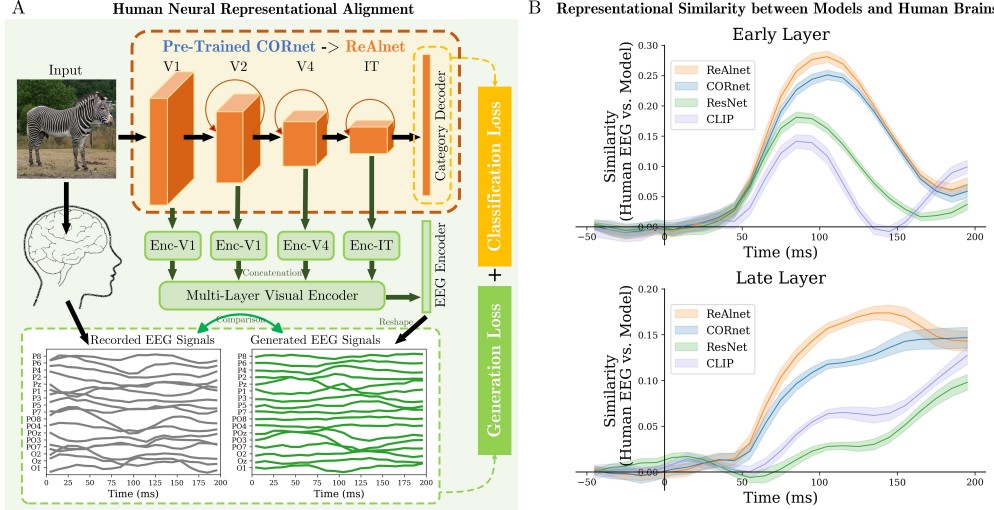

Figure 1: ReAlnet aligned with human neural signals as a more human brain-like vision model. (A) An overview of ReAlnet alignment framework. Adding an additional multi-layer encoding module to an ImageNet pre-trained CORnet-S, the outputs contain the category classification results and the generated EEG signals. Using training EEG data, we aim to minimize both *classification loss* and *generation loss*, enabling CORnet to not only stabilize the classification performance but also effectively learn human brain features and transform into ReAlnet. (B) Using test EEG data, we measure the representational similarity between the model RDM and timepoint-by-timepoint EEG neural RDMs for early and late layers in ReAlnet, CORnet-S, ResNet-101, and CLIP (with a ResNet-101 backbone) respectively (early layer: the first layer; late layer: the layer before the classification layer in ReAlnet, CORnet, and ResNet, and the last visual layer in CLIP), and ReAlnet shows the highest similarity to the human brain.

the exploration of new methodologies. Researchers have attempted various strategies, including *altering the model's architecture* (adding recurrent structures Kar et al. (2019); Kietzmann et al. (2019); Kubilius et al. (2019); Spoerer et al. (2017); Tang et al. (2018), dual-pathway models Bai et al. (2017); Choi et al. (2023); Han & Sereno (2022; 2023); Sun et al. (2017), topographic constraints Finzi et al. (2022); Lee et al. (2020); Lu et al. (2023); Margalit et al. (2023) or feedback pathways Konkle & Alvarez (2023) ) and *changing the training task* (using self-supervised training Konkle & Alvarez (2022); Prince et al. (2023) or 3D task models O'Connell et al. (2023)). However, limited studies have focused on directly using neural responses to complex visual information as feedback to improve the model's similarity to human brains. Our research focuses on a third approach – *utilizing human brain neural activity data to realize brain-like models*. This approach represents a more direct alignment strategy between models and the human brain, unconstrained by variations in model structure or pre-training methods, potentially marking a crucial step towards achieving greater resemblance to the human brain. Thus, our central research question emerges: **Can we use human brain activity to align ANNs on object recognition and achieve more human brain-like vision models?**

**Related Work.** Several previous studies have already started to try to apply neural data to machine learning especially deep learning models. The earliest attempt was to apply human fMRI signals to amend the classification boundary of SVMs and CNNs to achieve better category classification performance Fong et al. (2018). Some more recent studies started to let the models learn neural representations. One common way is to add a similarity loss to increase the representational similarity between models and neural activity (neural recordings from mouse V1, monkey V1 or IT) during the training Dapello et al. (2023); Federer et al. (2020); Li et al. (2019); Pirlot et al. (2022). Another strategy from Safarani et al. (2021) is to add an additional task based on an encoding module to predict monkey V1 neural activity. Both similarity-based method and multi-task framework can achieve more brain-like representations and improve model robustness. However, these neural align-

ment studies have two key challenges: (a) *Dependence on animal instead of human neural activity.* This limits the direct applicability and relevance of findings to human visual processing, and it is harder to enable models to effectively learn the human brain's representational patterns based on the low data quality. (b) *Single brain region or single model layer alignment.* On the one hand, previous studies could only align a single early or late layer in CNN and/or align the model with a certain brain region, V1 or IT. On the other hand, it remains unclear which specific brain region should align with which particular layer of the model, leading to potential misalignment and inaccuracies.

Additionally, a recent study focused on video emotion recognition first applied a representational similarity-based method to align CNN with human fMRI activity Fu et al. (2023). However, it is noteworthy that they focused on simpler emotion recognition tasks, may fall short in the more complex and diverse domain of object recognition which has larger space and multitude of object categories. Therefore, our work addressed this by employing an additional encoding module that goes beyond mere similarity. This module predicts human neural activity and is trained to autonomously extract complex visual features, offering a more effective approach for aligning the model with human neural representations in object recognition.

**Contributions.** To bridge the gap between AI vision and human vision, we propose a more human brain-like vision model, ReAlnet, effectively aligned with human brain representations based on a novel and effective encoding-based multi-layer alignment framework. We summarize our contributions and novel findings as follows:

- To the best of our knowledge, we are the first to directly align object recognition models using non-invasive neural data recorded from human brains, which opens new possibilities for enhancing brain-like representations in models based on human brain activity.

- We propose a novel image-to-brain encoding-based representational alignment framework that optimizes multiple layers of the network simultaneously, which effectively improves the model's similarity to human brain representations across different modalities (both human EEG and fMRI).

- Our representational alignment framework allows us to obtain a personalized vision model by aligning with individual's neural data.

- Aligning with human neural representations can improve the model adversarial robustness.

## 2 RE(-PRESENTATIONAL)AL(-IGNMENT)NET

Here we describe the human neural data (EEG data for the alignment, and both EEG and fMRI data for testing the similarity between models and human brains) we used in this study, the alignment pipeline (including the structure, the loss functions, and training and test methods) for aligning CORnet representations with human neural representations to obtain ReAlnet, and the evaluation methods for measuring representational similarity between models and human brains and adversarial robustness.

### 2.1 HUMAN EEG DATA FOR REPRESENTATIONAL ALIGNMENT

Human EEG data were obtained from an EEG open dataset, THINGS EEG2 Gifford et al. (2022), including EEG data from 10 healthy human subjects in a rapid serial visual presentation (RSVP) paradigm. Stimuli were images sized $500 \times 500$ pixels from THINGS dataset Hebart et al. (2019), which consists of images of objects on a natural background from 1854 different object concepts. Before imputing the images to the model, we reshaped image sizes to $224 \times 224$ pixels and normalized the pixel values of images to ImageNet statistics. Subjects viewed one image per trial (100ms). Each participant completed 66160 training set trials (1654 object concepts $\times$ 10 images per concept $\times$ 4 trials per image) and 16000 test set trials (200 object concepts $\times$ 1 image per concept $\times$ 80 trials). EEG data were collected using a 64-channel EASYCAP and a BrainVision actiCHamp amplifier. We use already pre-processed data from 17 channels (O1, Oz, O2, PO7, PO3, POz, PO4, PO8, P7, P5, P3, P1, Pz, P2) overlying occipital and parietal cortex. We re-epoched EEG data ranging from stimulus onset to 200ms after onset with a sample frequency of 100Hz. Thus, the shape of our EEG data matrix for each trial is 17 channels $\times$ 20 time points. and we reshaped the EEG data as a vector including 340 values for each trial. Before the model training and test, we averaged

all the repeated trials (4 trials per image in the training set and 80 trials per image in the test set) to obtain more stable EEG signals.

It is worth noting that the training and test sets do not overlap in terms of object categories (concepts), which means that the performance of ReAlnet trained on the training set, when evaluated on the test set, can effectively reveal the model's generalization capability across different object categories.

## 2.2 HUMAN fMRI DATA FOR CROSS-MODALITY TESTING

To demonstrate that our approach of aligning with human EEG not only enhances the model's similarity to human EEG but indicates that ReAlnet has effectively learned the human brain's representational patterns more broadly, we also performed cross-modal testing, testing ReAlnet on data from a different modality (fMRI), from a different set of subjects, viewing a different set of images. The fMRI data originate from Shen et al. (2019). This *Shen fMRI dataset* recorded human brain fMRI signals from three subjects while they focused on the center of the screen viewing natural images sourced from ImageNet. We selected he test set from *Shen fMRI dataset*, which comprises fMRI signals of each subject viewing 50 images of different categories, with each image being viewed 24 times. We averaged the fMRI signals across the 24 repeated trials to obtain more stable brain activity for each image observation and extracted signals from five regions-of-interest (ROIs) for subsequent comparison of model and human fMRI similarity: V1, V2, V3, V4, and the lateral occipital complex (LOC).

## 2.3 IMAGE-TO-BRAIN ENCODING-BASED ALIGNMENT PIPELINE

**Basic architecture of ReAlnet:** We have chosen the state-of-the-art CORnet-S model Kubilius et al. (2018; 2019) as the foundational architecture for ReAlnet, incorporating recurrent connections akin to those in the biological visual system and proven to more closely emulate the brain's visual processing.

**EEG generation module:** In addition to the original recurrent CNN structure, we have added an EEG generation module designed to construct an image-to-brain encoding model for generating realistic human EEG signals. Each visual layer is connected to a nonlinear $N \times 128$ layer-encoder (Enc-V1, Enc-V2, Enc-V4, and Enc-IT correspond to Layer V1, V2, V4, and IT) that processes through a fully connected network with a ReLU activation. These four layer-encoders are then directly concatenated to form an $N \times 512$ Multi-Layer Visual Encoder, which is subsequently connected to an $N \times 340$ EEG encoder through a linear layer to generate the predicted EEG signals. Here $N$ is the batch size.

Therefore, we aim for the model to not only perform the object classification task but also to generate human EEG signals which can be highly similar to the real EEG signals when a person views the certain image through the EEG generation module with a series of encoders. During this process of generating brain activity, ReAlnet's visual layers are poised to effectively extract features more aligned with neural representations.

**Alignment Loss:** Accordingly, the training loss $\mathcal{L}^A$ of our alignment framework consists of two primary losses, a classification loss and a generation loss with a parameter $\beta$ that determines the relative weighting:

$$\mathcal{L}^A = \mathcal{L}^C + \beta \cdot \mathcal{L}^G \tag{1}$$

$\mathcal{L}^C$ represents the standard categorical cross entropy loss for model predictions on ImageNet labels:

$$\mathcal{L}^C = -\sum_{i=1}^{N} y_i log(p_i) \tag{2}$$

Here, $y_i$ represents the $i$-th image, and $p_i$ represents the probability that model predicts the $i$-th image belongs to class $i$ out of 1000 categories. However, the correct ImageNet category labels for images in THINGS dataset are not available. Therefore, we adopt the same strategy as in Dapello et al.

(2023), using the labels obtained from the ImageNet pre-trained CORnet without neural alignment as the true labels to stabilize the classification performance of ReAlnet.

$\mathcal{L}^G$ is the generation loss, which includes a mean squared error (MSE) loss $\mathcal{L}^{MSE}$ and a contrastive loss $\mathcal{L}^{Cont}$ between the generated and real EEG signals. This contrastive loss is calculated based on the dissimilarity (one minus Spearman correlation coefficient) between generated and real signals, aiming to bring the generated signals from the same image (positive pairs) closer to the corresponding real human EEG signals and make the generated signals from different images (negative pairs) more distinct. $\mathcal{L}^G$ is calculated as followed:

$$\mathcal{L}^G = \mathcal{L}^{MSE} + \mathcal{L}^{Cont} \tag{3}$$

$$\mathcal{L}^{MSE} = \frac{1}{N} \sum_{i=1}^{N} (S_i - \hat{S}_i)^2 \tag{4}$$

$$\mathcal{L}^{Cont} = \frac{1}{N} \sum_{i=1}^{N} [1 - \rho(S_i, \hat{S}_i)] - \frac{1}{N(N-1)} \sum_{i=1}^{N} \sum_{j=1, j \neq i}^{N} [1 - \rho(S_i, \hat{S}_j)] \tag{5}$$

Here, $S_i$ and $\hat{S}_i$ represent the generated and real EEG signals corresponding to the $i$-th image.

**Training procedures:** Unlike CORnet which trained on the same ImageNet dataset, ReAlnet additionally trained on individual EEG data consists of 10 personalized ReAlnets, 1 per EEG subjects. Each network were trained to minimize the alignment loss including both classification and generation losses with a static training rate of 0.00002 for 30 epochs using the Adam optimizer. We used a batch size of 16, meaning the contrastive loss computed dissimilarities of 256 pairs for each gradient step. Also, we trained various ReAlnets using four different $\beta$ weights ($\beta = 1, 10, 100, 1000$) separately. In total, we trained 40 ReAlnets (4 $\beta$ weights $\times$ 10 subjects).

**Representational similarity analysis (RSA):** RSA is used for representational comparisons between models and human brains Kriegeskorte et al. (2008) based on first computing representational dissimilarity matrices (RDMs) for models and human neural signals, and then calculating Spearman correlation coefficients between RDMs from two systems.

To evaluate the similarity between models and human EEG, the shape of each RDM is $200 \times 200$, corresponding to 200 images in THINGS EEG2 test set. For EEG RDMs, we applied decoding accuracy between two image conditions as the dissimilarity index to construct EEG RDM for each timepoint and each subject. For model RDMs, we input 200 images into each model and obtained latent features from each visual layer. Then, we constructed each layer's RDM by calculating the dissimilarity using one minus Pearson correlation coefficient between flattened vectors of latent features corresponding to any two images. To compare the representations, we calculated the Spearman correlation coefficient as the similarity index between layer-by-layer model RDMs and timepoint-by-timepoint neural EEG RDMs. To evaluate the similarity between models and human fMRI, the shape of each RDM is $50 \times 50$, corresponding to 50 images in Shen fMRI dataset test set. For fMRI RDMs, we calculated one minus Pearson correlation coefficient between voxel-wise activation patterns corresponding to any two images as the dissimilarity index in the RDM for each ROI and each subject. For model RDMs, similar to the EEG comparisons above, we obtained the $50 \times 50$ RDM for each layer from each model. Then, we calculated the Spearman correlation coefficient as the similarity index between layer-by-layer model RDMs and neural fMRI RDMs for different ROIs, assigning the final similarity for a certain brain region as the highest similarity result across model layers due to the lack of a clear correspondence between different model layers and brain regions. All RSA analyses were implemented based on NeuroRA toolbox Lu & Ku (2020).

**Adversarial attacks:** For performing white box adversarial attacks, we used Fast Gradient Sign Attack (FGSA). We evaluated top-5 classification accuracies on ImageNet with epsilon ranged from 0 to 0.06 for each model.

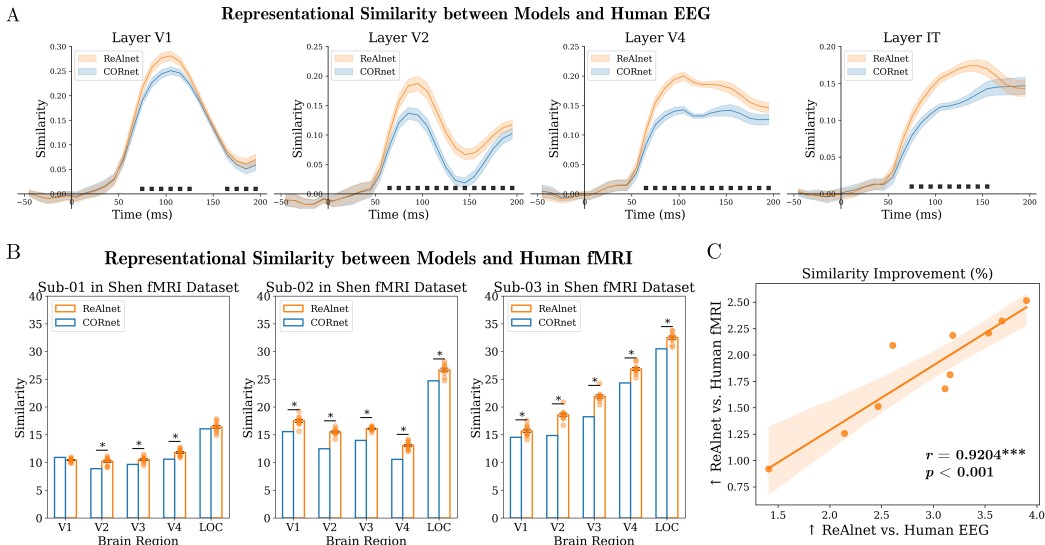

Figure 2: ReAlnets show higher similarity to human EEG and fMRI representations. (A) Representational similarity time courses between human EEG and models for different layers respectively. Black square dots at the bottom indicate the timepoints where ReAlnet vs. CORnet were significantly different ($p < .05$). Shaded area reflects ±SEM. (B) Representational similarity between three subjects' fMRI activity of five different brain regions and models respectively. Asterisks indicate significantly higher similarity of ReAlnet than that of CORnet ($p < .05$). (C) Correlation of similarity improvement between ReAlnet vs. human EEG and ReAlnet vs. human fMRI. Each circle dot indicates an individual ReAlnet.

## 3 RESULTS

### 3.1 IMPROVED SIMILARITY IN REALNETS TO HUMAN EEG

Here, for each of the 10 human subjects, we calculated (1) the similarity between their EEG data and the single CORnet, and (2) the similarity between their EEG data and the subject-matched ReAlnet. ReAlnets show significantly higher similarity to human EEG neural dynamics for all four visual layers (Layer V1: 70-130ms and 160-200ms; Layer V2: 60-200ms; Layer V4: 60-200ms; Layer IT: 70-160ms) than the original CORnet without human neural alignment (Figure2A). Further statistical analysis of each layer's similarity improvement (ReAlnet - CORnet) and improvement ratio ((ReAlnet - CORnet) / CORnet) also indicate that at the similarity peak timepoint, there is a maximum of an 8% similarity improvement and an 80% improvement ratio, with the average improvement for the 50-200ms time-window being over 5% and the average improvement ratio over 40% (Figure2B). Additional comparisons also show that ReAlnet is more human brain-like than not only CORnet, but also ResNet and CLIP (Figure1B).

These results suggest three findings: (1) Our multi-layer alignment framework indeed improves all layers' similarity to human EEG representations. (2) Every ReAlnet with individual neural alignment exhibits improved similarity to human EEG compared to the basic CORnet. (3) ReAlnets demonstrate the generalization of improvement in human brain-like similarity cross object categories, as the image categories used for testing were entirely absent during the alignment training.

### 3.2 IMPROVED SIMILARITY IN REALNETS TO HUMAN FMRI

Although ReAlnet demonstrates higher similarity to human EEG, a question arises: Does ReAlnet learn representations specific to EEG, or more general neural representations of the human brain? To ensure that our alignment framework enables the model to learn representations beyond the single

modality of EEG, we utilized additional human fMRI data to evaluate the model's cross-modality representational similarity to human fMRI.

Excitingly, we indeed observed an increase in this cross-modal brain-like similarity. The results indicate that even though ReAlnets were aligned based on human EEG data, they still resemble the human brain more closely on fMRI data compared to CORnet (Figure2B). Also, there is a significant correlation of ReAlnets' similarity improvement compared to CORnet between EEG and fMRI ($r$=.9204, $p < .001$) (Figure2C).

These findings further highlight three points: (1) Across multiple ROIs, ReAlnets exhibits higher human fMRI similarity than CORnet. (2) Despite not being trained with the EEG data of subjects in the fMRI dataset, almost every ReAlnet shows higher fMRI similarity, suggesting that ReAlnet learns consistent brain information processing patterns across subjects. (3) Images from fMRI dataset for evaluation were never presented during the alignment training, reaffirming the generalization of ReAlnets in improving brain-like similarity across object categories and images.

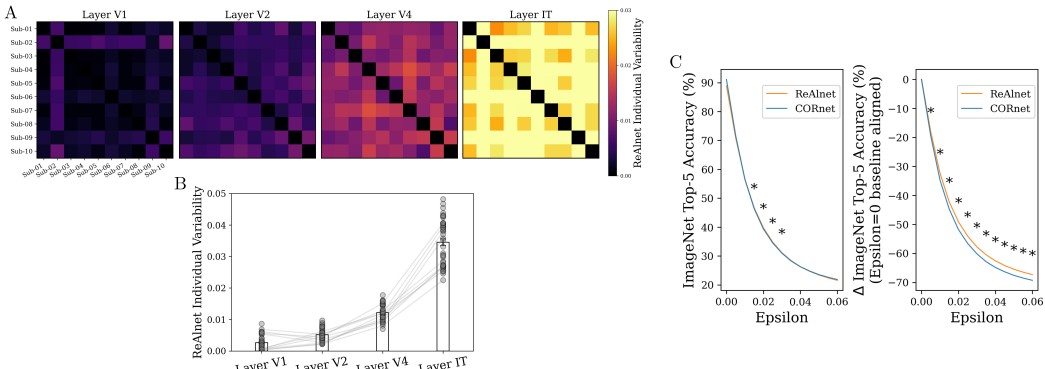

Figure 3: Individual variability across ten personalized ReAlnets and increased adversarial robustness in ReAlnets. (A) ReAlnet individual variability matrices of four visual layers. (B) ReAlnet individual variability along the model layers. Each circle dot indicates a pair of two different ReAlnets. (C) Original (left) and baseline-aligned (right) adversarial robustness for ReAlnets and CORnet as a function of Epsilon. Asterisks indicate significantly higher adversarial robustness of ReAlnets than that of CORnet ($p < .05$).

### 3.3 Hierarchical individual variabilities in ReAlnets

Unlike traditional models in computer vision, ReAlnet is a personalized model trained based on different individual's neural data. This sparked our interest in exploring whether these personalized ReAlnets exhibit intra-model individual variabilities and how such variabilities change across different layers of the model. To investigate this, we conducted comparisons between model RDMs based on 200 images in THINGS EEG2 test set across different layers, using the dissimilarity (one minus the Spearman correlation coefficient) between two RDMs corresponding to two ReAlnets as an individual variability index.

Our results show: (1) Personalized ReAlnets indeed exhibit individual variability (Figure3A and Figure3B). (2) This variability increases with the depth of the layers (from Layer V1 to Layer IT, Figure3A and Figure3B). This may also suggest a trend of increasing individual variability from primary to higher visual cortical areas in human brains.

### 3.4 Increased adversarial robustness in ReAlnets

Using white-box FGSA, we also discovered that ReAlnets, aligned with human neural representations, have increased adversarial robustness against adversarial attacks (Figure3C). The left panel of Figure3C shows a slight increase in adversarial robustness in ReAlnets compared to CORnet at around Epsilon = 0.02. However, the original classification performance (Epsilon = 0) of ReAlnets

is lower than that of CORnet, due to the absence of correct labels for images in THINGS EEG2 dataset. To make a fairer comparison, we aligned the classification accuracy at Epsilon = 0 as the baseline to observe the relative decline in accuracy for both ReAlnets and CORnet as Epsilon value increases. The corrected results (Figure3C right) demonstrate more pronounced adversarial stability in ReAlnets.

## 3.5 ReAlnet performance across different weights

The results presented above are based on a generation loss weight $\beta$ set to 100. We further explored the impact of this $\beta$ value on the performance of ReAlnet. Theoretically, a higher $\beta$ should lead to stronger learning of human neural representations. However, is a larger $\beta$ always better? Our findings suggest otherwise.

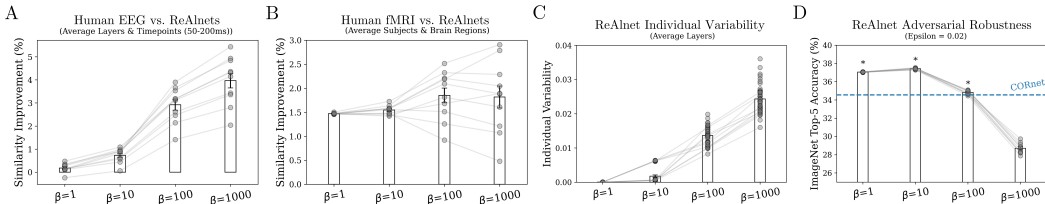

Figure 4: ReAlnet performance across different $\beta$ values. (A) Improvement in human EEG similarity of ReAlnets compared to CORnet (averaging four visual layers and timepoints in a 50-200ms time-winodw). (B) Improvement in human fMRI similarity of ReAlnets compared to CORnet (averaging three subjects and five brain regions). (C) ReAlnet individual variability (averaging four visual layers). (D) Adversarial robustness of ReAlnets and CORnet when Epsilon = 0.02. Asterisks indicate significantly higher adversarial robustness of ReAlnets than that of CORnet ($p < .05$).

We observed that with an increase in $\beta$, ReAlnets show greater similarity to human EEG and fMRI (Figure4A and Figure4B) and more pronounced individual variability within models (Figure4C). However, an increase in $\beta$ also reduces the improvement of adversarial stability (the improvement at $\beta = 100$ was less significant than at $\beta = 1$ or 10) (Figure4D). Moreover, at excessively high values ($\beta = 1000$), ReAlnet's adversarial robustness was even lower than the original CORnet without neural alignment (Figure4D). Therefore, this analysis suggests that: (1) It justifies our use of $\beta = 100$ as a weight that balances the trade-offs and maximizes advantages of ReAlnet. (2) $\beta$ is a parameter that could be manipulated differently in future research depending on research goals.

## 3.6 Control experiments

For the control experiments, we tested two aspects: (1) How does contrastive learning influence model-to-brain alignment? (2) If we disrupt the pairing of each image with the EEG signal from the same subject but elicited by viewing a different image, can the model still learn the neural representation patterns of the human brain? Accordingly, we trained two additional sets of ReAlnets based on human EEG data from ten subjects in THINGS EEG2 dataset, termed as *W/o ContLoss* models (without the constrastive loss component) and Unpaired models (where the pairing between images and EEG signals was disrupted).

The results of the control experiments reveal: (1) *W/o ContLoss* models still exhibit an improvement in human brain similarity compared to CORnet. However, while the similarity to human EEG did not decrease compared to ReAlnet, the similarity to cross-modality human fMRI significantly decreased. This suggests that the contrastive loss component in our alignment framework aids ReAlnet in extracting more cross-modality brain visual representation features. (2) Unpaired models failed to enhance brain similarity, which show no significant improvement in brain similarity compared to CORnet, indicating that the training process requires the model to effectively learn the specific neural visual features corresponding to each image. Only in this way can the model become more human brain-like and then exhibit higher similarity to the human brain across different object images, categories, and human neuroimaging data modalities.

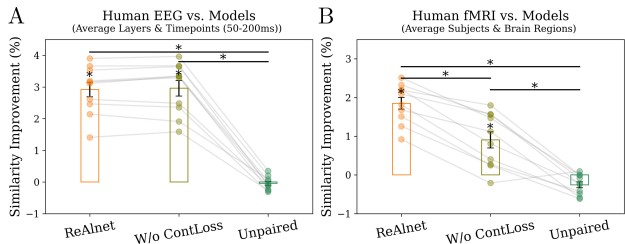

Figure 5: Results of control experiments. (A) Improvement in human EEG similarity of ReAlnets and control models compared to CORnet. (B) Improvement in human fMRI similarity of ReAlnets and control models compared to CORnet. Asterisks indicate the significance ($p < .05$).

## 4 DISCUSSION

Building upon previous research utilizing neural data for aligning object recognition models, we have proposed a novel and more effective framework for human neural representational alignment, along with the corresponding human brain-like model, ReAlnet. Unlike previous studies that focused on using animal neural signals to optimize models or were unable to use global neural activity for comprehensive model optimization Dapello et al. (2023); Federer et al. (2020); Li et al. (2019); Pirlot et al. (2022); Safarani et al. (2021), our approach efficiently utilizes human neural activity to simultaneously optimize multiple layers of the model, enabling it to learn the human brain's internal representational patterns for object visual processing. Notably, unlike prior research relying on behavioral or single modality neural recording data for model evaluation Dapello et al. (2023); Federer et al. (2020); Fu et al. (2023); Li et al. (2019); Pirlot et al. (2022); Safarani et al. (2021), we employed different modalities of human neuroimaging data for model evaluation to ensure that ReAlnet learns broader, cross-modal brain representational patterns. Additionally, we observed that ReAlnet exhibits individual representational variabilities akin to human brain's hierarchical processing and adversarial stability similar to the findings in other brain-inspired models Dapello et al. (2023); Konkle & Alvarez (2023).

Regarding ReAlnet itself, it warrants further exploration to ascertain what specific information has learned from the alignment with human brains. The fact that different generation loss weights do not significantly impact the behavioral performance but do enhance its similarity to human brains suggests that nodes in the model, which originally did not encode category-specific information, may have been optimized Federer et al. (2020). More analyses of the neural network's internal representations may be needed to delve into this. Also, from a reverse-engineering perspective, attempting to understand the brain-like optimization process of the model could further aid in unraveling the mechanisms by which our brains process visual information Ayzenberg et al. (2023); Cic (2019); Doerig et al. (2023); Kanwisher et al. (2023); Lu & Ku (2023); Lu & Golomb (2023b).

Certainly, it is important to highlight that ReAlnet transcends being merely a specific vision model; it represents a pioneering framework potentially applicable for aligning any AI model with brain activity. On the one hand, this alignment framework can be extended to other neural modalities, such as fMRI and MEG (dimensionality reduction might be necessary for extensive neural data features), paving the way for the development of variants like ReAlnet-fMRI and ReAlnet-MEG. On the other hand, the ambition is to adapt this framework to a wider range of models and tasks in the future, including language and auditory processing and self-supervised or unsupervised models, leading to innovations such as ReAlnet-Language, ReAlnet-Auditory, and self-supervised or unsupervised versions of ReAlnet.

## 5 CONCLUSION

Our study transcends traditional boundaries by employing a groundbreaking alignment framework that pioneers the use of human neural data to achieving a more human brain-like vision model, ReAlnet. Demonstrating significant advances in bio-inspired AI, ReAlnet not only aligns closely with

human EEG and fMRI but also exhibit hierarchical individual variabilities and increased adversarial robustness, mirroring human visual processing. We hope that our alignment framework stands as a testament to the potential synergy between computational neuroscience and machine learning and enables the enhancement of any AI model to be more human brain-like, opening up exciting possibilities for future research in brain-like AI systems.

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

## A APPENDIX

### A.1. ImageNet classification performances of ReAlnets at different $\beta$ values

We tested the classification accuracy of ReAlnets on ImageNet at different $\beta$ values (Figure 8). Importantly, to ascertain that the observed decrease in accuracy was not due to the additional generation task compromising classification performance, but rather the absence of correct ImageNet labels for images in THINGS EEG2 dataset, we trained a ReAlnet with $\beta = 0$. This ReAlnet excluded the EEG signal generation module but underwent fine-tuning with images from THINGS EEG2 dataset. The results indicated that the ReAlnet with $\beta = 0$ also experienced a similar level of decline.

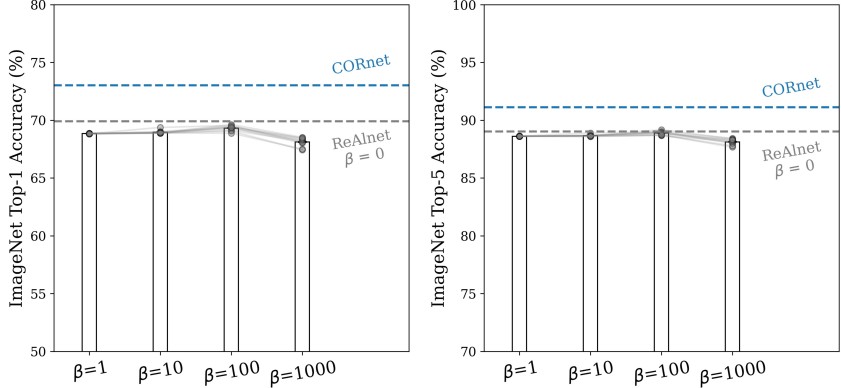

Figure 6: ImageNet classification accuracy of different ReAlnets. Left: Top-1 accuracy. Right: Top-5 accuracy. The blue dotted line indicates the accuracy of CORnet, and the grey dotted line indicates the accuracy of ReAlnet at $\beta = 0$.

## A.2. EEG generation performances of our alignment framework at different $\beta$ values

We evaluated the EEG generation performance of the alignment frameworks at different ( values by calculating the Spearman correlation between the generative EEG signals and the actual EEG signals. Figure 7 shows the EEG generation performance and some examples of generated results.

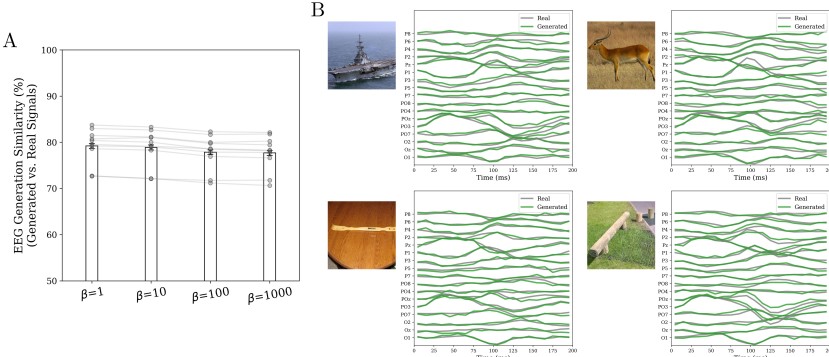

Figure 7: (A) EEG generation performance of different alignment frameworks. (B) Four examples of EEG generation results (from the model at $\beta = 100$ of Sub-01). For each example, the left image indicates the image input to the ReAlnet and the image viewed by the subject. The grey curves represent the real EEG signals, and the green curves represent the generated EEG signals corresponding to the same image.

### A.3. Representational similarity between human EEG and ReAlnets at different $\beta$ values

Figure 8 shows the representational similarity between human EEG and ReAlnets at different $\beta$ values.

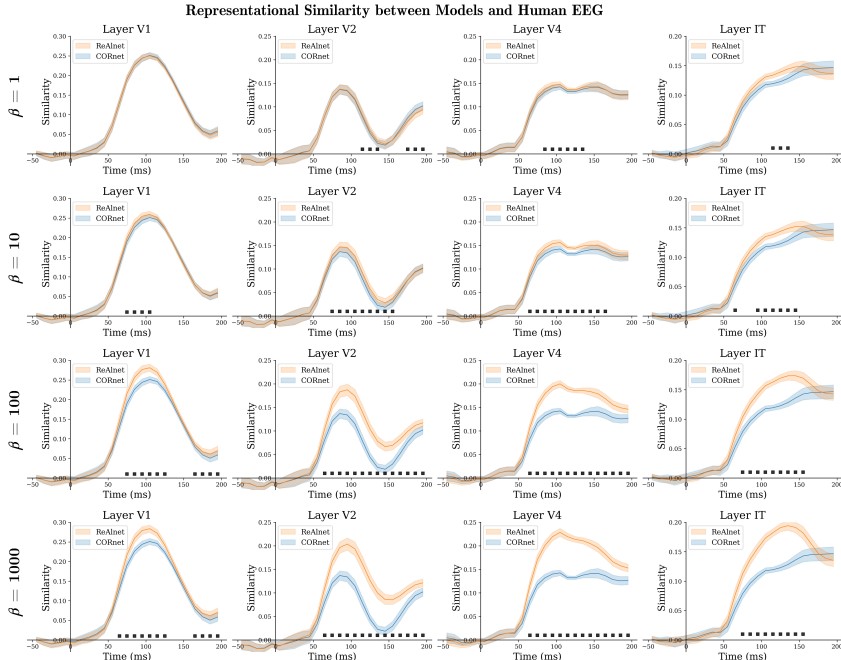

Figure 8: Representational similarity time courses between human EEG and different ReAlnets for different layers respectively. Black square dots at the bottom indicate significant timepoints ($p < .05$). Shaded area reflects ±SEM.

### A.4. Representational similarity between human fMRI and ReAlnets at different $\beta$ values

Figure 9 shows the representational similarity between human fMRI and ReAlnets at different $\beta$ values.

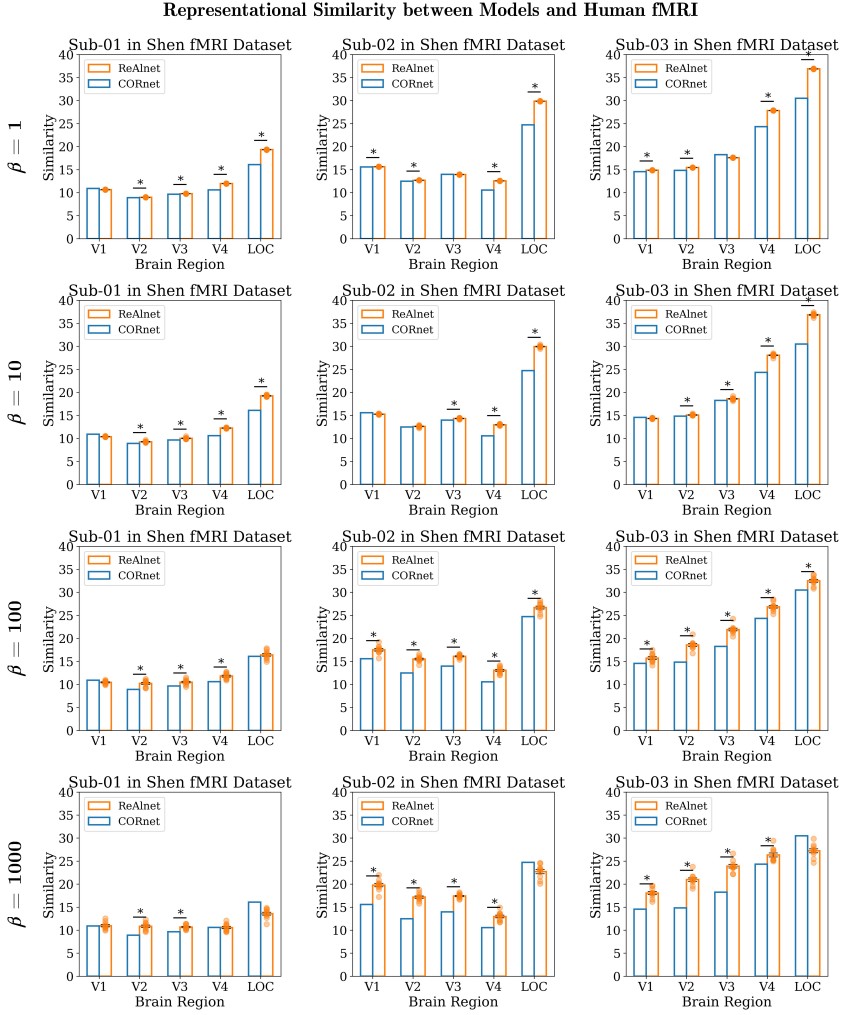

Figure 9: Representational similarity between three subjects' fMRI activity of five different brain regions and different ReAlnets respectively. Asterisks indicate significantly higher similarity of ReAlnet than that of CORnet ($p < .05$).

### A.5. Individual variability across personalized ReAlnets at different $\beta$ values

Figure 10 shows the representational similarity between human fMRI and ReAlnets at different $\beta$ values.

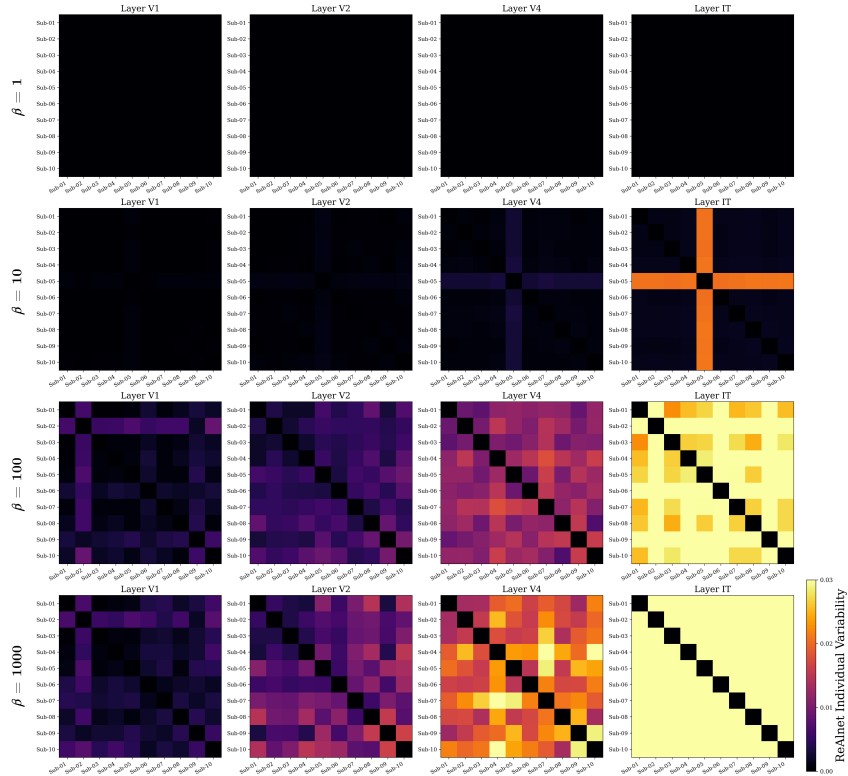

Figure 10: Individual variability matrices of four visual layers of different ReAlnets.

### A.6. Adversarial robustness of ReAlnets at different $\beta$ values

Figure 11 shows the adversarial robustness of ReAlnets at different $\beta$ values.

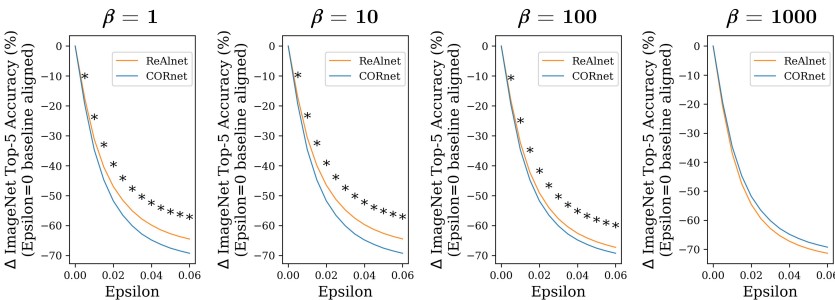

Figure 11: Baseline-aligned adversarial robustness for different ReAlnets as a function of Epsilon. Asterisks indicate significantly higher adversarial robustness of ReAlnets than that of CORnet ($p < .05$).

### A.7. Representational similarity between human brains and controlled models

Figure 12A shows the representational similarity between human EEG and controlled models, and Figure 12B shows the representational similarity between human fMRI and controlled models.

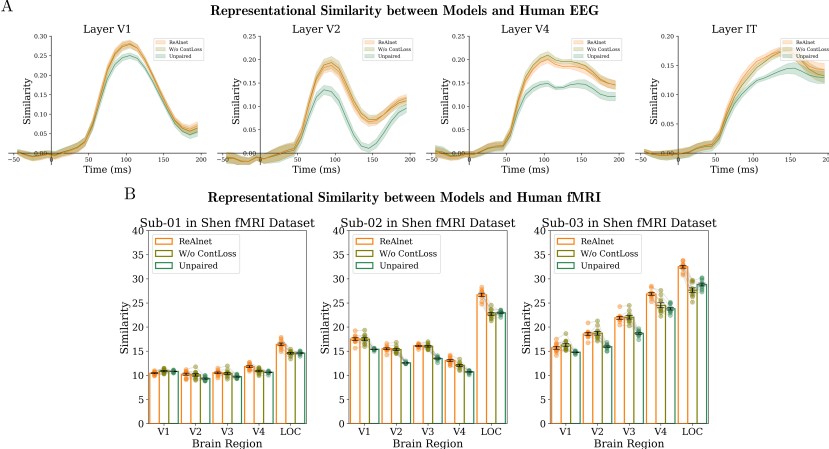

Figure 12: (A) Representational similarity time courses between human EEG and ReAlnets and control models ($\beta$ for different layers respectively. Shaded area reflects±SEM. (B) Representational similarity between three subjects' fMRI activity of five different brain regions and ReAlnets and control models ($\beta = 100$) respectively.

