# OpenReview forum: "ReAlnet: Achieving More Human Brain-Like Vision via Human Neural Representational Alignment"
_ICLR.cc/2024/Workshop/Re-Align — ICLR 2024 Workshop Re-Align Poster_

### Official Review · Reviewer_TKXo · 2024-02-19

**Rating:** 3
**Fit:** 3
**Confidence:** 2

**Workshop Review:**

**Summary:** \
In the study, the authors propose a new vision model called ReAlnet consisting of a CORNet-S backbone and adding an EGG generation model. By using an alignment loss that not only optimizes for classification prediction but also EEG signal prediction they show that while reaching a slightly less ImageNet prediction accuracy compared to vanilla CORNet-S an improved similarity to human EEG signal even generalizing to fMRI signal without being trained for it. This ReAlnet exhibits additional robustness against white-box (FGSA) adversarial attacks.

---

**Clarity:** good \
**Correctness:** good \
**Novelty:** good \
**Interest to the community:** good

---

**Strong Points:**
- **Very well-written** manuscript which facilitates the understanding of the work and results.
- **Good clarity:** The aim of the paper is clear. The ReAlnet offers a great contribution to further understanding which visual features make models more aligned with the human visual system. The alignment loss seems to be an innovative and interesting way to make DCCNs more aligned with human brain measurements.
- **Great figures**, visualizing the various results clearly and comprehensively.
- The research done here seems timely, and relevant for various fields - including neuroscience to better understand the processing of the visual system for object recognition; Human-AI alignment improving the alignment between comp. models and brain signals, and computer vision by creating more robust vision models.

---

**Questions:**
Question 1: It is a cool finding that the EEG signal fine-tuned model also generalizes to alignment with fMRI measurements. Would you expect similar results when adding an fMRI activity prediction unit and testing on EEG?
Question 2: Why does introducing a ß (1, 10) result in a drop in Top1/Top5 ImageNet accuracy, and why is it higher for ß =100 and then drops again?

**Reason For Not Giving Higher Score:**

N/A

**Reason For Not Giving Lower Score:**

N/A

**Reviewer Domain:**

cognitive science

---

### Official Review · Reviewer_v7sV · 2024-02-23
**Incremental, but highly relevant work**

**Rating:** 3
**Fit:** 3
**Confidence:** 1

**Workshop Review:**

Clarity 4/5
Correctness 5/5
Novelty 4/5
Interest 4/5

The authors propose -- ReAlnet -- a framework for aligning convolutional neural network representations with human EEG and fMRI signals to achieve more human brain-like vision models.

Their experiments show representational similarity to both EEG and fMRI after alignment and increased adversarial robustness.

**Reason For Not Giving Higher Score:**

Everything felt like an incremental contribution

**Reason For Not Giving Lower Score:**

Manuscript was well written and quite thorough

**Reviewer Domain:**

machine learning

---

### Official Review · Reviewer_nguT · 2024-02-25
**A promising paper that fits the workshop well, but embedding in literature and claims of robustness can be improved**

**Rating:** 2
**Fit:** 3
**Confidence:** 2

**Workshop Review:**

- Clarity: the paper’s aims and claims are very clear. Some details of the methods require elaboration.
- Correctness: the results largely appear correct, some of them require additional controls and raise new questions.
- Novelty: the paper claims to be the first to directly align object recognition models using non-invasive human brain data. I think this is correct, although the paper could benefit from more careful embedding in related literature on end-to-end training on neural data and EEG encoding/decoding (see comments below).
- Interest to the community: This paper is highly likely to be of interest to the workshop.

**Reason For Not Giving Higher Score:**

- It is not hugely surprising that adding a loss to improve similarity with EEG results in better similarity with EEG (even on a test set). Prior work has trained DNNs end-to-end on non-invasive brain measurements (e.g. Gifford et al., used here; Seeliger & van Gerven), and used them to decode images from fMRI. Early works show that EEG signal can be used to decode images using very simple image metrics (e.g. Ghebreab et al., 2009, NeurIPS). So, non-invasive brain signals contain a lot of information about images; thus it makes sense that pretrained DNN representations can be mapped to and used to generate predicted samples of these signals. The paper would benefit from a stronger embedding in this encoding/decoding literature, and more in-depth discussion of visual information present in EEG signals.
- The novelty lies in the claimed benefits: 1) generalization to fMRI, and 2) increased adversarial robustness. The initial results here are promising but both will benefit from more comprehensive testing; for 1) the reported increase on the Shen fMRI dataset appear robust for all three subjects, but I’m not sure I understand the (incredibly high!?) correlation of this increase for the 10 subject-specific ReAINets on the (subject-averaged?) fMRI improvements (Fig 2C); what is driving this correlation? 2) Robustness is explored in just one experiment, for which the choice and settings are not motivated; a more comprehensive test on a clear benchmark is necessary.
- An important aspect is if the alignment affects performance on the main task (here object recognition), but this is now only reported in the Appendix, showing that the ReAInets performance is lowered. This should be reported and discussed up front. Similarly, the ReAInet reported in Appendix A1 (trained with the same labels but excluding the EEG generation module) is an important and possibly more appropriate baseline model than the original ImageNet trained CorNet and should feature in the main results.

**Reason For Not Giving Lower Score:**

- Demonstrating increased robustness of on object recognition through representational alignment with non-invasive human data is very interesting.
- The generalisation to a second non-invasive modality (fMRI) is impressive.
- The simplicity of the approach (1 non-linear+ 1 linear mapping of 4 layer network to downsampled and vectorized EEG) is attractive.
- The paper already includes several important controls (training on unpaired data, training without contrastive loss, training with beta set to 0).
- The paper raises a lot of interesting questions for further analyses, e.g. what aspects of the CorNet representations change when adding the EEG-based loss, why some subject’s aligned ReAINets generalize better to fMRI than others, why such networks show increased adversarial robustness, etc.‘

**Reviewer Domain:**

neuroscience

---

### Decision · Program_Chairs · 2024-03-02

**Decision:**

Accept (Poster)

**Comment:**

We invite this work for presentation as a poster at the workshop. However, we strongly recommend that the authors temper and properly scope the title of the paper, as the evidence presented is insufficient to justify broad and advanced claims of "achieving human brain-like vision" with the methods presented here (for which analogous methods have been proposed before and many ongoing developments will be presented at this workshop), and the notion of "human neural representational alignment" employed only covers a single method for representational alignment (for which there are many alternative methods; see Sucholutsky et al. (2023)).